# Ring-closing C−O/C−O metathesis of ethers with primary aliphatic alcohols

Hongmei Liu[1], Qing Huang[1], Rong-zhen Liao [1], Man Li[1] ✉ & Youwei Xie [1] ✉

In canonical organic chemistry textbooks, the widely adopted mechanism for the classic transetherifications between ethers and alcohols starts with the activation of the ether in order to weaken the C−O bond, followed by the nucleophilic attack by the alcohol hydroxy group, resulting in a net C−O/O−H σ-bond metathesis. In this manuscript, our experimental and computational investigation of a $Re_2O_7$ mediated ring-closing transetherification challenges the fundamental tenets of the traditional transetherification mechanism. Instead of ether activation, the alternative activation of the hydroxy group followed by nucleophilic attack of ether is realized by commercially available $Re_2O_7$ through the formation of perrhenate ester intermediate in hexafluoroisopropanol (HFIP), which results in an unusual C−O/C−O σ-bond metathesis. Due to the preference for the activation of alcohol rather than ether, this intramolecular transetherification reaction is therefore suitable for substrates bearing multiple ether moieties, unparalleled by any previous methods.

Ethers are ubiquitous motifs commonly seen in a wide range of solvents, fragrances, disinfectants, herbicides, drug intermediates, fuels, lubricants, precursors for polymers, etc[1–3]. The canonical Williamson ether synthesis (Fig. 1a) is still a method of choice for the preparation of unsymmetrical ethers nowadays because of its operational simplicity and general applicability, despite the necessity for the stoichiometric amount of bases and environmentally hazardous alkyl halides[4]. Numerous well-known strategies have been developed to complement this classical methodology, including Mitsunobu-type etherification reactions[5], reductive deoxygenation of esters[6], hydroetherification of alkenes[7], oxidative etherification of aryl C−H bonds[8], transition-metal-catalyzed Ullmann-type synthesis[9], etc. Transetherification (Fig. 1b) between an easily available ether and an alcohol provides a promising alternative to unsymmetrical ethers by direct exchange of "RO−" in the inert functionality of "RO−R"[10], a reaction that has been frequently applied in total syntheses[11], materials science[12], polymer chemistry[13], biomass conversion[14], etc. However, it suffers from limited scope due to the sole reaction mode available for both aliphatic (Fig. 1c) and aromatic ethers (Fig. 1d) via acid catalysis[10,15–19], transition-metal-catalyzed reversible cleavage of C(sp²)−O bonds[20], as well as recently burgeoning techniques such as electrocatalysis[21] and photocatalysis[14]. These methodologies proceed exclusively through ether activation,

resulting in a formal C−O/O−H σ-bond metathesis. Alternatively, activation of the alcohol followed by an OH departure facilitated by the nucleophilic attack of ether (Fig. 1e) will result in an unusual C−O/C−O σ-bond metathesis, a strategy potentially capable of solving the inherent limitation associated with traditional transetherification. However, to the best of our knowledge, this reaction mode is elusive even for an intramolecular variant due to the difficulty of selective OH activation in the presence of ether moieties, as well as the low nucleophilicity of ether, which is incapable of displacing an OH group[22,23], making its displacement by a second OH group more likely to deliver dehydrative coupling of two alcohols[18,24].

The ring-closing metathesis reactions of alkenes[25] and alkynes[26] have evolved into one of the methods of choice in assembling cyclic compounds of varying ring sizes (Fig. 1f), which are of irrebuttable significance in both organic synthesis[27] and materials science[28]. Recently, with the advent of novel multiple-bond metathesis reactions such as those between alkenes and carbonyl compounds[29–31], this strategy has been empowered with more flexibility regarding substrate preparation (Fig. 1g). On the other hand, much less effort has been dedicated to the development of metathesis reactions of the more easily available C−C and C−X bonds[32–37], the intramolecular variant of which could potentially provide rapid access to saturated

[1]Hubei Key Laboratory of Bioinorganic Chemistry and Materia Medica, School of Chemistry and Chemical Engineering, Huazhong University of Science and Technology, Wuhan, China. ✉e-mail: manli_hx@hust.edu.cn; xieyw@hust.edu.cn

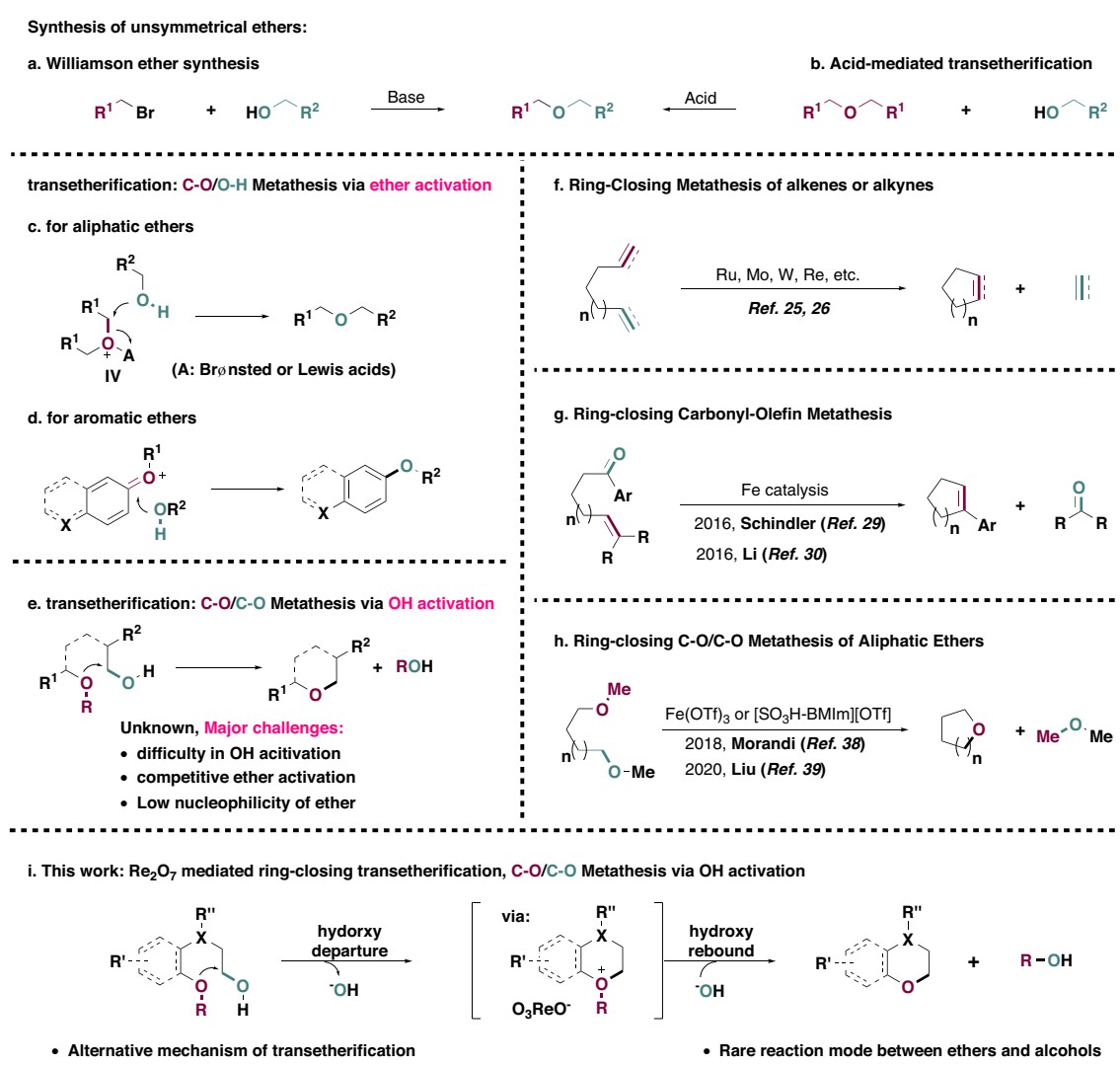

**Fig. 1 | Synthesis of unsymmetrical ethers and transetherification reactions.**
**a** Williamson ether synthesis. **b** Acid mediated transetherification reactions.
**c** Mechanism of acid mediated transetherification for aliphatic ethers. **d** Mechanism of transetherification for aromatic ethers. **e** Transetherification of ethers and
alcohols via OH activation, C–O/C–O σ-bond metathesis. **f** Ring-closing alkene
metathesis. **g** Ring-closing carbonyl-olefin metathesis. **h** Ring-closing C–O/C–O σ-
bond metathesis of aliphatic ethers. **i** Ring-closing C–O/C–O σ-bond metathesis via
OH activation, a mechanism reversal in transetherification.

carbo- and heterocyclic compounds. In 2018, the Morandi group
reported a novel Fe(OTf)$_3$ mediated ring-closing C−O/C−O σ-bond
metathesis of aliphatic ethers for the synthesis of cyclic ethers[38].
Recently, the Liu group used an ionic liquid under metal-free
conditions[39] and the Kerton group utilized functionalized biochar[40],
respectively, to catalyze the same transformation (Fig. 1h).

Here we show a Re$_2$O$_7$ mediated ring-closing σ-bond metathesis
between ethers and aliphatic alcohols. The success of this protocol
relies on selective hydroxy group activation via perrhenate ester
formation[41,42], even in the presence of acid-labile ether or acetal
functionalities[43], which provides an unusual C−O/C−O σ-bond
metathesis reaction between ethers and aliphatic alcohols that repre-
sents a reversal in the mechanism of transetherification. This method
can provide facile access to various substituted chromanes and several
other frequently encountered cyclic ethers.

## Results
### Reaction optimization
To test this hypothesis, we started to evaluate the title reaction with an
easily accessible substrate **1a** that bears a primary alcohol and an aryl

methyl ether moiety (Table 1). Excitingly, both Re$_2$O$_7$ and HReO$_4$
effectively promoted the C−O/C−O σ-bond metathesis reaction at
100 °C to give 20−30% yields (entries 1−2). In contrast, other forms of
rhenium catalysts were ineffective (entries 3−5). Moreover, some
conventional Lewis acids that were successfully applied in previous
C−O/C−O σ-bond metathesis of aliphatic ethers[38] or C−O/O−H σ-bond
metathesis between alcohols and ethers were also tested (entries
6−10). However, only Fe(OTf)$_3$ and Bi(OTf)$_3$ (Supplementary Table 1)
gave trace amounts of the cyclization products with same catalyst
loadings. We next studied the influence of different solvents on the
performance of Re$_2$O$_7$. After extensive screening, it was found that
fluorinated alcohol significantly improved the yield (entries 11−12), and
HFIP[44−46] gave a good yield of 86%. More polar solvents such as DMF,
DMSO, HMPA, acetonitrile etc. were detrimental to the reaction and no
product were formed when they were used with Re$_2$O$_7$ (Supplemen-
tary Table 1). Further increasing the catalyst loading or reaction tem-
perature did not lead to improved chemical yields, while lowering the
catalyst loading or reaction temperature significantly reduced the
reaction efficiency (entries 13−14), some common Brønsted and Lewis
acids were also tested, however, they showed either no catalytic ability

## Table 1 | Optimization of the reaction condition[a]

| Entry | Catalyst | Loading | Solvent | T (°C) | [1]H-NMR yield |
|---|---|---|---|---|---|
| 1 | $Re_2O_7$ | 2% | DCE | 100 | 30% |
| 2 | $HReO_4$ | 4% | DCE | 100 | 24% |
| 3 | $MeReO_3(MTO)$ | 2% | DCE | 100 | NR |
| 4 | $(CO)_5ReBr$ | 2% | DCE | 100 | NR |
| 5 | $Re(CO)_{10}$ | 2% | DCE | 100 | NR |
| 6 | $Fe(OTf)_3$ | 2% | DCE | 100 | Trace |
| 7 | $CuCl_2$ | 2% | DCE | 100 | NR |
| 8 | $AlCl_3$ | 2% | DCE | 100 | NR |
| 9 | $FeCl_3$ | 2% | DCE | 100 | NR |
| 10 | $V_2O_5$ | 2% | DCE | 100 | NR |
| 11 | $Re_2O_7$ | 2% | $CF_3CH_2OH$ | 100 | 54% |
| 12 | $Re_2O_7$ | 2% | HFIP | 100 | 86% (85%)[b] |
| 13 | $Re_2O_7$ | 1% | HFIP | 100 | 70% |
| 14 | $Re_2O_7$ | 2% | HFIP | 80 | 69% |
| 15 | $Fe(OTf)_3$ | 2% | HFIP | 100 | 67% |

[a]Reactions were performed with **1a** (1.0 equiv.), solvent (0.2 M), catalyst (1–5 mol%) at 100 °C for 12 hours. [b]isolated yield.

or significantly reduced efficiency, even with HFIP as the solvent (see Supplementary Table 1 for a complete list of reaction conditions screened). Triflate salts gave moderate yields when HFIP was applied as the solvent (entry 15), however, further [18]O labeling study showed that the product was a mixture of C–O/C–O and C–O/O–H metathesis products for aliphatic ethers (Supplementary Fig. 16).

### Synthetic scope

With the optimized reaction condition for the ring-closing C–O/C–O σ-bond metathesis in hand, we next explored the substrate scope of this transformation (Fig. 2, upper half). Generally, methyl groups at various positions of the aromatic ring are all well tolerated (**2b**–**2e**), even for the one with a methyl group at the *ortho* position to the ether moiety (**2b**), which gave reduced yield due to the increased steric hindrance. Substrates bearing a methoxy group (**2f**), fluorine (**2g**), chlorine (**2h**), and bromine (**2i**) atoms all reacted under the standard condition, providing products in good yields with a handle for further elaboration. Substrate with a phenolic OH could potentially undergo either oxidation or alkylation under the reaction condition, surprisingly, product resulting from the ring-closing C–O/C–O σ-bond metathesis could still be obtained in moderate yield (**2j**). Substrates bearing polyaryl alkyl ethers were less soluble. Fortunately, however, these reactions still proceeded to give the products in good yields (**2k**–**2m**). The substrate with a labile 2-thiophenyl group (**2n**) also gave the ring-closing C–O/C–O σ-bond metathesis product in good yield, with slightly higher catalyst loading (3 mol%) at a higher reaction temperature (120 °C). Acetoxyl group (**2o**) and a carbonyl group (**2p**) can have intricate hydrogen-bonding interactions with multiple HFIP molecules and therefore decrease the nucleophilicity of the ether group, leading to significantly reduced chemical yields. When electron-withdrawing groups were not directly attached to the aromatic ether, good yields could still be obtained (**2q**, **2r**). The substituent at the γ-position to the hydroxy group was well tolerated (**2 s**) while that at the β-position (**2t**) resulted in reduced yield (50%) due to the formation of rearranged side products. Finally, an interesting double

ring-closing C–O/C–O σ-bond metathesis gave tricyclic compound **2 u** in 38% yield, alongside with significant amount of alkylated intermediate that could not further transform to desired product. Encouraged by these preliminary results, we then tried to expand the reaction scope to alcohols bearing an aliphatic ether moiety (Fig. 2, Lower half). To our delight, the optimized reaction condition for aromatic ethers can be applied to aliphatic ethers without any modification, and reactions proceeded smoothly to provide tetra-hydrofuran (**2v**), tetrahydropyran (**2w**) and oxepane (**2x**) in good yields. Larger cyclic ether could not be synthesized by this method and only recycled starting material was obtained (**2y**). Synthesis of biologically significant heterocycles such as morpholines (**2z**) and dioxanes (**2aa**) via ring-closing C–O/C–O σ-bond metathesis turned out to be challenging in previous reports due to the strong coordinating ability of these substrates, requiring the application of a large amount of catalysts (20–50 mol%) and prolonged reaction time[38,39]. However, this was not an issue in our case and the reaction could still be realized with a small amount of catalysts (4–5 mol%), probably due to the fact that the new reaction mode relied on activation of the hydroxy group rather than the ether activation as in the case of Lewis-acid-catalyzed transformations, lending further support to the advantage of this method. Phenyl (**2bb**) substituent at the THP rings was well tolerated. Methyl (**2cc**) and methoxy (**2dd**) groups on the aryl group were beneficial to the reaction, and substrates bearing halogen-substituted aryl groups (**2ee**–**2gg**), multiply substituted aryl groups (**2hh**–**2ii**) all reacted to deliver the corresponding products in good to excellent yields. In contrast to aromatic ethers, electron-withdrawing groups such as trifluoromethyl (**2jj**), carbonyl (**2kk**), phosphonyl (**2ll**) were all tolerated, although in some cases increased catalysts loading and reaction temperature were necessary to achieve a good conversion. The substrates with a 2-thiophenyl group (**2 mm**) or polyarene substituents (**2nn**–**2pp**) all reacted also gave the ring-closing C–O/C–O σ-bond metathesis product in good to excellent yields under the standard reaction condition. Substrates with secondary, tertiary alcohols or allylic alcohols easily ionized under $Re_2O_7$ catalysis to give mixtures of dehydration or

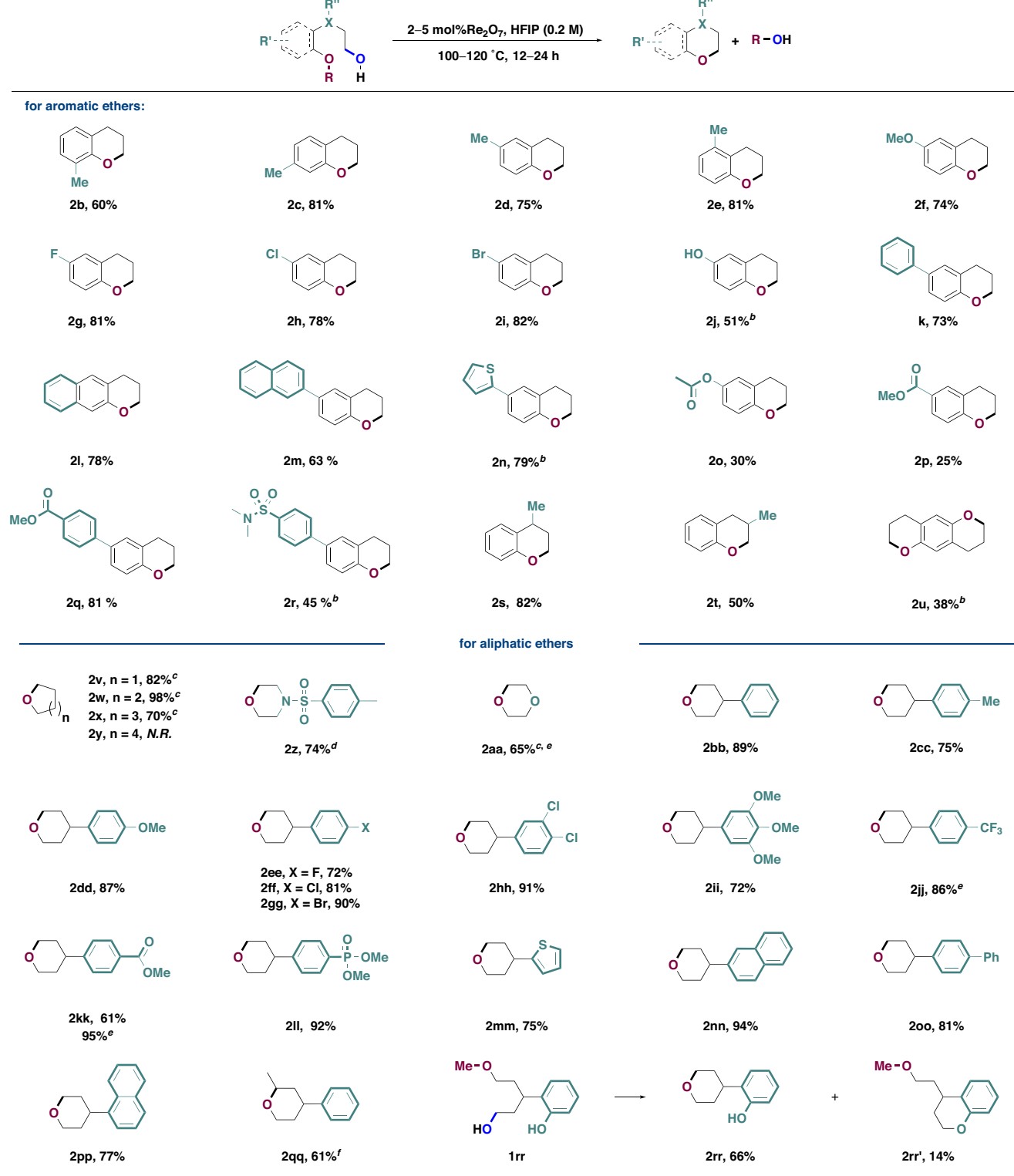

**Fig. 2 | Ring-closing C–O/C–O σ-bond metathesis of ethers with alcohols.**
[a]Reaction conditions: reactions were performed with **1** (1.0 equiv.), solvent (0.2 M), catalyst (2 mol%) at 100 °C for 12 h, **i**solated yields were provided unless mentioned otherwise. [b]3 mol% Re₂O₇, 120 °C, 24 h. [c]Yields were determined by analyzing ¹H NMR of the reaction mixture with an internal standard. [d]4 mol% Re₂O₇, 120 °C, 24 h. [e]5 mol% Re₂O₇, 120 °C, 24 h. [f]2 mol% Re₂O₇, 80 °C, 12 h.

Friedel-Crafts alkylation products. However, secondary alkyl group could be placed on the ether part to deliver the product in good yield (**2qq**). When a substrate bears both aliphatic OH and phenolic OH, the C–O/C–O σ-bond metathesis between aliphatic OH and ether dominates (**2rr**) along with small amount of product resulting from condensation of the two hydroxy groups (**2rr′**).

## Discussion

As shown in Fig. 3 (pathway **a**, pink arrow) the initial design of our method involved the direct activation of the hydroxy group via the formation of perrhenate ester **I** from substrate **A**[41]. Intramolecular S$_N$2 type attack of the ether oxygen on the perrhenate ester results in C–O bond cleavage to yield an oxonium ion intermediate **II**, a process

**Fig. 3 | Plausible reaction pathway. Hydroxy activation. a** Hydroxy activation via perrhenate ester formation (pink arrow). **b** Hydroxy activation via Brønsted acid catalysis (blue arrow). **Ether activation: c** Activation of aliphatic ether by Brønsted acid (orange arrow). **d** Activation of aromatic ether by Brønsted acid (purple arrow).

facilitated by two HFIPs that stabilize the $ReO_4^-$ in the form of $[ReO_4(HFIP)_2]^-$[42]. $[ReO_4(HFIP)_2]^-$ then attacks **II** from the methyl group to give the desired product **B** and methyl perrhenate ($MeOReO_3$), which undergoes transesterification with substrate **A** to release methanol and regenerate the intermediate **I**, entering the following catalytic cycle. Alternatively, the key intermediate **II** can also be formed via direct protonation of the hydroxy group (**III**) followed by an intramolecular $S_N2$ type attack of the ether oxygen on the protonated hydroxy group (Fig. 3, pathway **b**, blue arrow). Moreover, an ether activation mechanism might also be operative, giving C−O/O−H σ-bond metathesis rather than C−O/C−O σ-bond metathesis. For aliphatic ethers (Fig. 3, pathway **c**, orange arrow), the protonation of ether (or by Lewis acid activation: not shown) provides oxonium intermediate **V**, followed by an intramolecular displacement by the hydroxy group to generate the intermediate **VI**, which delivers the product **B** after deprotonation. This mechanism is similar to the previous ring-closing C−O/C−O σ-bond metathesis with slight adjustment to meet the situation in this work[38]. For aromatic ethers (Fig. 3, pathway **d**, purple arrow), protonation of the electron-rich anisole moiety provides the reactive key intermediate **VII**. An intramolecular nucleophilic addition-and-elimination sequence gives the desired product **B** and regenerates the catalyst, a process similar to previously reported C−O/O−H σ-bond metathesis reactions between aromatic ethers and aliphatic alcohols[10,17–19]. It is impossible to confirm one pathway while completely ruling out the other three based solely on the reaction outcome, since all pathways result in the same products and by-products.

## Results and discussion
### Mechanistic investigations
We then carried out a series of control experiments to possibly reveal the true mechanism underlying this ring-closing σ-bond metathesis reaction. (Fig. 4) Firstly, we methylated the hydroxy groups in both substrates **1a** and **1bb** to give **1a'** and **1bb'**. Unlike the non-methylated substrates, **1a'** did not react to provide desired product in a significant amount while **1bb'** reacted extremely sluggish to give desired **2bb** in only 12% NMR yield (Fig. 4a). Substrate with phenolic OH and aliphatic ether (Supplementary Fig. 8) only give the cyclization product in 15%

NMR yield that resulted from activation of aliphatic ether followed by attack of the phenolic either (confirmed by [18]O labelling experiments, Supplementary Fig. 16). A more interesting substrate **3** was also tested, in which both traditional C−O/C−O σ-bond metathesis of two ethers and cross C−O/C−O σ-bond metathesis between ethers and alcohols can occur, unsurprisingly, the latter metathesis reaction was predominant to deliver product **4** in 75% yield (Fig. 4b). These results, which were in sharp contrast to previously reported C−O/C−O σ-bond metathesis between two ethers[38–40], confirmed the critical role of the hydroxy group and pointed to an activation of the hydroxy group via perrhenate ester formation rather than acid catalysis, because otherwise the protonation of the methyl ether followed by intramolecular displacement of methanol would also lead to the formation of the key intermediate **II** (Fig. 3) should the later pathway be operative.

Next, we studied the influence of the alkyl group of the ether on the reaction outcome by taking the crude [1]H NMR immediately when the reaction was finished (Fig. 4c). For the methyl group (**1a**), **1a'** was detected in 14%, which was likely to result from an attack of **1a** on the oxonium intermediate **II**, methanol was detected while dimethyl ether (resulted from an attack of methanol on **II**) was not. However, we postulated that the absence of dimethyl ether might be due to its high volatility and the difficulty of detecting it with [1]H NMR. Indeed, for substrate with an aryl ethyl ether, diethyl ether was detected in small amounts as a minor side product (6%), and for substrates with aryl propyl and aryl butyl ethers, alcohols were detected as the predominant byproducts while dialkyl ethers were detected in only trace amounts. Moreover, alkylation of substrate (**1a'**) was also not significant for substrates with alkyl groups larger than an ethyl group. These results were in good agreement with the involvement of the key intermediate **II**: the steric hindrance increased with the size of the alkyl group, making the displacement by a second alcohol more difficult than by $[ReO_4(HFIP)_2]^-$, which is ion-pairing with **II**. The formation of alkylated substrate **1a'** also explained the relative lowered yields for substrates with aryl methyl ether and aryl ethyl ether (86% and 87% respectively) compared to substrates with aryl propyl ether and aryl butyl ether (98% and 96% respectively), since alkylated substrate **1a'** could not be converted to the desired product due to the absence of

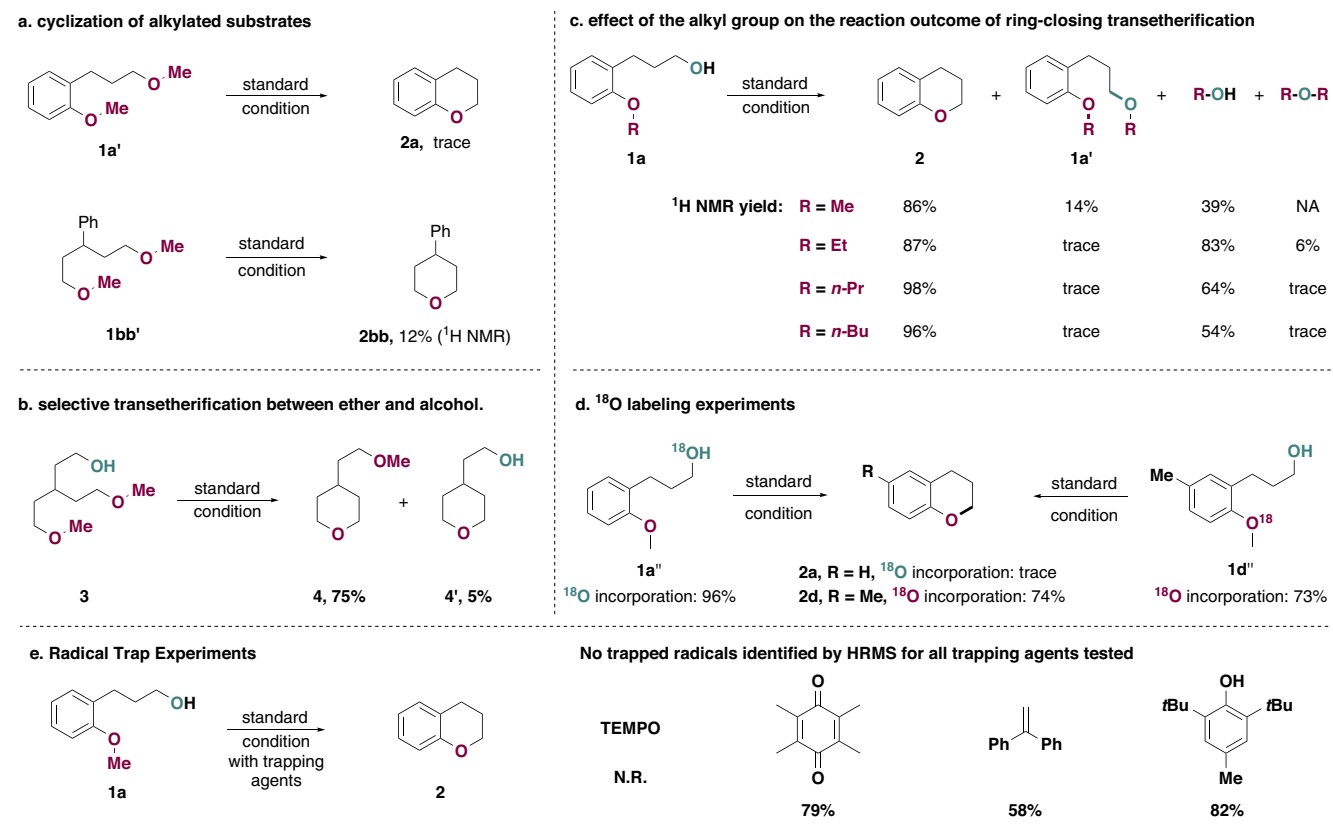

**Fig. 4 | Mechanistic experiments. a** Cyclization of alkylated substrates. **b** Cyclization of substrates with multiple ethers that can have manifold reaction modes. **c** Analyzing the yields of products and byproducts of the reaction by changing the alkyl groups of ether. **d** [18]O labeling experiments.

hydroxy group (Fig. 4a). Moreover, **1a"** with an [18]O labeled hydroxy group did not lead to any [18]O incorporation in the cyclization product, while **1d"** with an [18]O labeled ether moiety led to complete [18]O incorporation in the product, confirming that the oxygen atom originated from the ether moiety (Fig. 4d). [18]O incorporation experiments with aliphatic ethers also showed a preference for hydroxy activation (Supplementary Fig. 16). Finally, experiments with various radical trapping agents successfully delivered the product in good yields (Fig. 4e), except for TEMPO, the basicity of which is known to be incompatible with $Re_2O_7$. No trapped radicals could be detected for all trapping agents tested, ruling out the possibility of a radical pathway[47]. Collectively, these results are best explained by hydroxy group activation pathway via perrhenate formation for the ring-closing C−O/C−O σ-bond metathesis (Fig. 3, pathway **a**, pink arrow).

## DFT calculations

Parallel to experimental studies, density functional theory (DFT) calculations were conducted to provide further mechanistic insights into the title reaction. The ring-closing C−O/C−O σ-bond metathesis of **1a** was taken as the model reaction. Various reaction pathways involving the aromatic substrate (pathways **a**, **b**, and **d**) proposed in Fig. 3 were tested theoretically. The calculated results regarding pathways **a** and **b** are listed in Fig. 5 (for more details on the DFT calculations, see the supplementary information).

As shown for pathway **a** in Fig. 5, with $Re_2O_7$ as the initial catalyst, **1a** readily undergoes a reversible esterification process via a six-membered ring transition state **TS1**[43] to generate the perrhenate ester **Int1**. **TS1** has a total barrier of only 16.6 kcal/mol, and the formation of **Int1** is found to be exergonic by 3.6 kcal/mol, rendering **Int1** a possible resting state for the upcoming transformation. **TS1** assisted with one and two HFIPs have been considered and are found to have higher energy barriers perhaps due to the steric clash between the oxygen

atoms of $Re_2O_7$ and the fluorine atoms of HFIP (see the Supplementary Information 4.2 for details). At **Int1**, an intramolecular $S_N2$-like step via **TS2**, which involves the assistance of two HFIPs, enables the formation of the proposed oxonium ion intermediate **Int2** after releasing the $[ReO_4(HFIP)_2]^-$ complex[42]. The total barrier of **TS2** is calculated to be 22.6 kcal/mol relative to **Int1** plus $(HFIP)_2$, and the formation of **Int2** is exergonic by 7.7 kcal/mol. Next, $[ReO_4(HFIP)_2]^-$ attacks **Int2** at its methyl group via an $S_N2$-like transition state **TS3** with a total barrier of 24.3 kcal/mol (**Int1** was taken as the resting state), leading to the final product **2a** and the generation of methyl perrhenate $(MeReO_3)$, and this step is exergonic by 11.0 kcal/mol. Subsequently, $MeReO_3$ could undergo a transesterification (**TS4**, total energy barrier of 20.6 kcal/mol) with **1a** to regenerate **Int1** that enters the next catalytic cycle. Based on the above results, the rate-determining step of pathway **a** is **TS3** with a total barrier of 24.3 kcal/mol when taking **Int1** as the resting state, and such an energy barrier is able to be overcome under the given experimental condition. Alternatively, the formation of the oxonium ion **Int2** via a Brønsted acid catalysis mechanism (pathway **b**, red line) was found to undertake a concerted yet asynchronous transition state **TS1'** that involves multiple bonds formation and cleavage, and **TS1'** is calculated to be 3.6 kcal/mol less favored than **TS2** in pathway **a**. In addition, the total energy barrier of the aromatic ether activation pathway (pathway **d** in Fig. 3) is calculated to be 4.5 kcal/mol higher than that of the perrhenate ester formation pathway **a** (see the Supplementary Information 4.2 for more details). Besides, the generation of various radical intermediates was also considered but was found to be thermodynamically unfeasible, thus the radical mechanism was excluded (Supplementary Fig. 24). Taken together, the computational results support that pathway **a**, which involves the assistance of two HFIPs and the formation of perrhenate ester, is the most favored one. This is in good agreement with the experimental mechanistic insights revealed in Fig. 4.

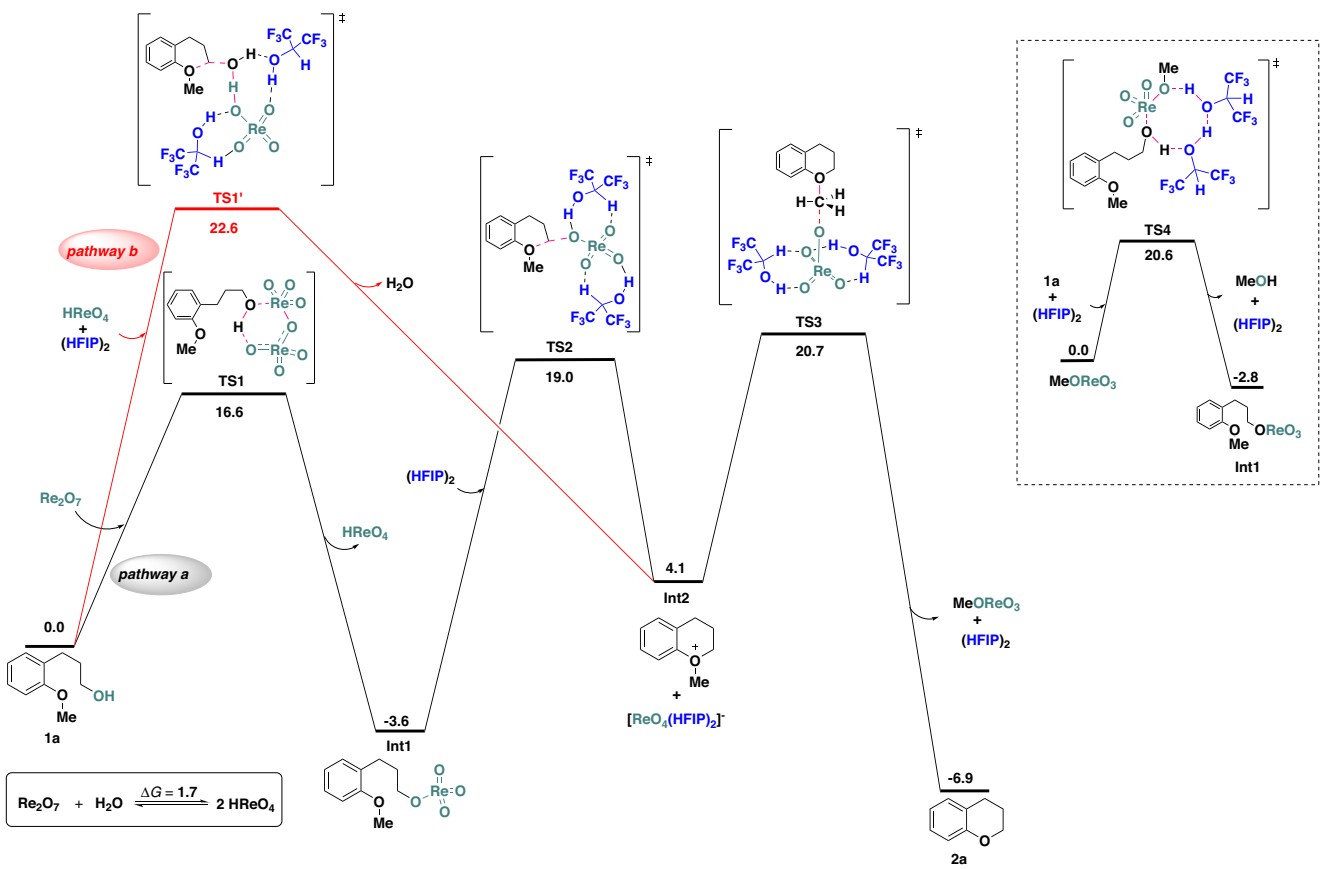

**Fig. 5 | Computed energy profile of the proposed mechanism.** Gibbs free energy profile (kcal/mol) at the SMD-B3LYP-D3/def2-TZVPP//B3LYP-D3/def2-SVP-SDD(Re) level for the ring-closing C–O/C–O σ-bond metathesis of **1a** catalyzed with $Re_2O_7$.

## Efficiency improvement based on the understanding of reaction mechanism

Inspired by these mechanistic studies, we speculated that it might be possible to enhance reaction efficiency by tuning the alkyl groups of the ether substrates in which the hydroxy group alkylation was responsible for the reduced chemical yield. We chose to explore substrate **1u** considering its double ring-closing C–O/C–O σ-bond metathesis delivered **2u** in only 38% isolated yield with the formation of a significant amount of alkylated substrates (**5**, **6**, **7**, **8**) by interim analysis of the reaction mixture with GC-MS. Excitingly, changing the two methyl groups to two propyl groups (**1u'**) retarded the substrate alkylation (only **5'** and **6'** were detected in smaller amounts) and increased the isolated yield of double ring-closing C–O/C–O σ-bond metathesis product significantly to 77% (88% per cyclization, Fig. 6a), it is worth to mention that only **5** and **5'** could further cyclize to give the desired product. On the other hand, changing the methyl group of **1b** to a propyl group (**1b'**) provided a less reactive substrate, giving a reduced yield of the desired product **2b** (Fig. 6b). This is because of the fact that steric hindrance rather than substrate alkylation was the major reason for lowered yield in the latter case, and changing the methyl group to a propyl group further increased the steric hindrance.

We have reported in this article the first ring-closing transetherification between ethers and alcohols. This method allows facile access to substituted THFs and THPs, as well as some other cyclic ethers. Mechanistic experiments, as well as DFT calculations, agreed with a reaction pathway in which the reaction is enabled by $Re_2O_7$ catalyzed hydroxy activation rather than ether activation in previously reported ring-closing C–O/C–O σ-bond metathesis of ethers. This reaction is also significantly different from other transetherification reactions between alcohols and ethers, which exclusively proceed via ether activation that result in C–O/O–H σ-bond metathesis. We anticipate that this methodology will enrich the toolbox for synthetic chemists for the construction of substituted THFs and THPs. From a broader perspective, we envisage that this reaction design provides valuable insight for developing novel cross-metathesis reactions involving inert $C(sp^3)$ –X as well as $C(sp^2)$ –X bonds.

## Methods

General procedure for the ring-closing C–O/C–O cross metathesis of ethers with primary aliphatic alcohols catalyzed by $Re_2O_7$ to prepare **2a**: To a 10 mL Schlenk tube was added 1.36 mg $Re_2O_7$ (2 mol%), 24.1 mg **1a** (1.0 equiv.) and 0.7 mL HFIP (0.2 M). The reaction mixture was stirred at 100 °C for 12 h, after which the reaction was quenched by adding 20 μL $Et_3N$, and the solvent was removed under reduced pressure. The crude mixture was then purified by flash column chromatography (100% petroleum ether) to afford the target product **2a** as a colorless oil (16.5 mg, 85% yield). Full experimental details, more details about the mechanistic studies, DFT calculations, and characterization of the newly described compounds can be found in the Supplementary Information.

## Data availability

The data that support the findings of this study are available within the article and the Supplementary Information. Details about materials and methods, experimental procedures, characterization data, mechanistic studies, DFT calculations and NMR spectra are available in the Supplementary Information. Cartesian coordinates of all the optimized structures are provided in Supplementary Data 1.

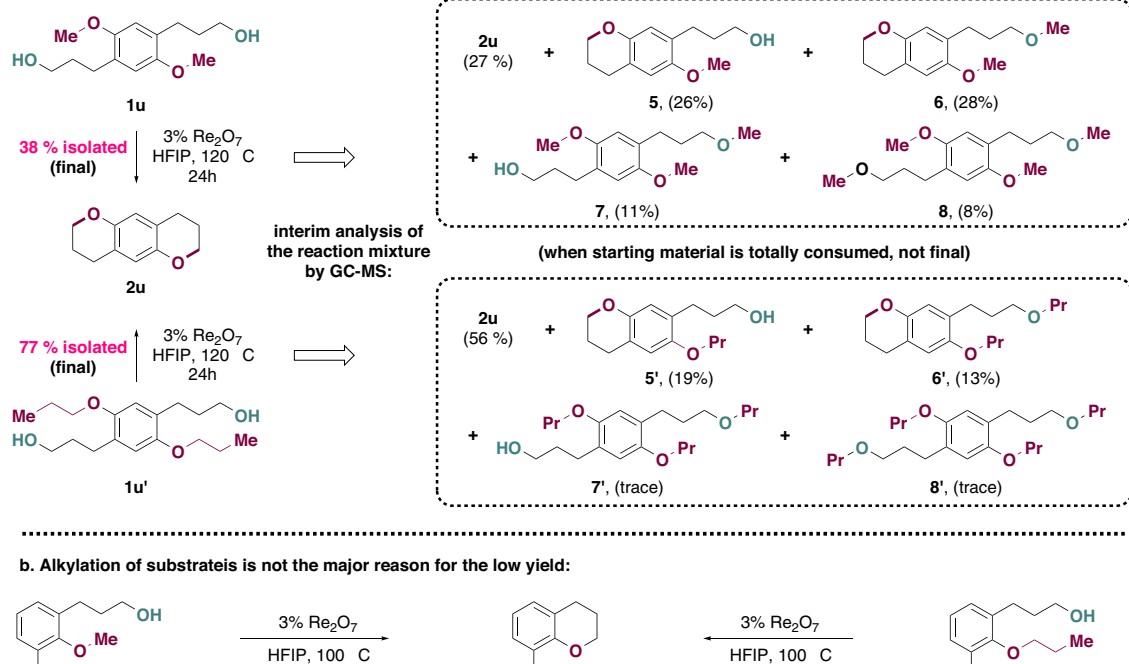

**a. Alkylation of substrate is the major reason for the low yield:**

**b. Alkylation of substrateis is not the major reason for the low yield:**

**Fig. 6 | Efficiency improvement by tuning the alkyl groups of ethers. a** Alkylation of substrate is the reason for low chemical yields. **b** Alkylation of substrate is not the reason for low chemical yields. Yields shown in parenthesis are determined by interim analysis of the reaction mixture with GC-MS when the starting material is totally consumed. **5** and **5′** could further cyclize to give the desired product, while **6** and **6′**, **7** and **7′**, **8** and **8′** are dead-end side-products.

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

## Acknowledgements

Financial support for this work is provided by the National Science Foundation of China (NSFC, 21801084 and 22171095), and Hubei Technological Innovation Project (2019ACA125). We thank B. List (Max-Planck-Institut für Kohlenforschung) and P. E. Floreancig (University of Pittsburgh) for insightful discussions during the implementation of this project. We are grateful to the Analytic and Testing Centre of HUST for data characterization.

## Author contributions

L.H. and X.Y. conceived the project. L.H., and H.Q. designed and performed the synthetic experiments. L.R.-Z., and L.M. carried out DFT calculations. L.H., H.Q., L.R.-Z., L.M., and X.Y. analyzed the data. L.H. and X.Y. wrote the manuscript.

## Competing interests

The authors declare no competing interests.
