## [Peer Review File · Nature Communications]

REVIEWER COMMENTS

Reviewer #1 (Remarks to the Author):

This manuscript by Xie et al. describes the development of a Re₂O₇ mediated ring-closing transesterification reactions between alcohols and ethers through a net C-O/C-O σ -bond metathesis to provide unsymmetrical ethers. This method allows selective access to various ether compounds, including substituted THFs, THPs, as well as cyclic ethers. A series of controlled experiments and DFT calculations were carried out. Based on the studies, a reversal of mechanism in transesterification for ring-closing C-O/C-O metathesis of ethers with primary aliphatic alcohols was proposed. The activation of the hydroxy group followed by nucleophilic attack of ether is realized by commercially available Re₂O₇ through the formation of perrhenate ester intermediate in hexafluoroisopropanol (HFIP), [ReO₄(HFIP)₂]⁻. The results are useful and valuable in synthetic chemistry and catalysis. In conclusion, this reviewer recommend acceptance for publication in the light of following comments.

1) For the optimization of the reaction conditions, the results of the applications of the Bronsted acids should be added in Table 1. In addition, the boiling point of hexafluoroisopropanol (HFIP) is 59 °C. However, the reactions were performed under 100-120 °C. How to perform the reaction? Is there a safety problem? Furthermore, HFIP plays crucial roles in this transformation. However, only Re₂O₇ catalyst was investigated in HFIP. Other catalysts should also be investigated in HFIP. These results are valuable for this chemistry and the readers.

2) For the scope of substrate, there are several issues should be addressed. A) Could secondary and tertiary alcohols be applied in this transformation? B) Functional group tolerance should be investigated, such as carbonyl, ester, amino, etc. C) Why is the yield of product 2p low?

3) In order to compare, the same reaction conditions should be carried out for Fig. 5A and Fig. 5B.

4) For Supporting Information, the authors should check the data carefully. For examples, the reported ¹H NMR data in the experimental details section does not match the [spectrum/reported HRMS formula] for compound(s): 1m; the reported ¹³C NMR data in the experimental details section does not match the [spectrum/reported HRMS formula] for compound(s): 1g, 1h, 1q, 1r; HRMS data of products 2a-2k, 2m-2cc and 2ee-2ff were not attached, and relevant data were missing; the data for these new compounds need to be collated in SI.

Reviewer #2 (Remarks to the Author):

In this manuscript, the authors report the Ring-closing C–O/C–O Metathesis of Ethers with Primary Aliphatic Alcohols leading to the synthesis of cyclic ethers. Similar transformations are already known for the synthesis of cyclic ethers via transesterification of two ether groups. However, this is the first example of cyclic ether synthesis via transesterification of alcohol and ether. But, the mechanistic investigation and anticipated novelty are insufficient for the proposed plausible mechanism and the authors should consider the following points:

- 1) C-O/C-O σ -bond metathesis strategy is already known for the synthesis of cyclic ethers that follows the activation of ethers and nucleophilic attack by another ether. But here authorS highlighted a different mechanism for metathesis that includes the activation of alcohol followed by a nucleophilic attack of less nucleophilic ethers which is new from the mechanistic point of view. Although it is the first transesterification between alcohol and ether, similar transformations are known.
- 2) Entry 6 of table 1 gave 10% yield in presence of DCE solvent so this reaction should be carried out in presence of fluorinated alcohol solvent.
- 3) Optimization of the reaction condition has been carried out in presence of solvent DCE, TFE and HFIP but no other polar solvents have been screened with Lewis acids.
- 4) No phenolic OH substrates have been screened in substrate scope. Is it possible to activate phenolic OH? What about the further possible nucleophilic attack by ether?.
- 5) In substrate scope (Table-2) what will happen if allylic alcohols are used?. Will it undergo further cyclization?
- 6) In mechanistic investigation (Figure-2) author has not given any evidence for the formation of perrhenate ester I.
- 7) In mechanistic investigation (Figure-2) no separate experiment has been carried out for the identification of oxonium ion intermediate II and intermediate VI.
- 8) The evidences in support of intermediates for plausible mechanism (Figure-2) are insufficient.
- 9) The possible involvement of radical and other intermediates can not be ruled out.
- 10) Evidence for O-H activation (Figure-3a, 3b) is enough to conclude that reaction is initiated via O-H activation but what will happen if one tries to carry out the reaction with dual OH activation containing both aliphatic OH groups or one aliphatic and another phenolic OH group?
- 11) How the author was able to get 18O incorporated substrate 1a'' and 1d''? A preparation procedure should be provided.
- 12) In (Figure-3c) of entry1, 1a' was observed in 14% yield but the authors have not mentioned anything regarding this product in the optimization table. Maybe this is also a possible intermediate in a reaction that facilitates the C-O/C-O σ -bond metathesis.

As a result, this reviewer finds this manuscript not suitable for publication in Nature Communications.

RESPONSES TO THE REVIEWER'S COMMENTS:

Reviewer #1:

- 1) **Comment:** (1) For the optimization of the reaction conditions, the results of the applications of the Bronsted acids should be added in Table 1. In addition, the boiling point of hexafluoroisopropanol (HFIP) is 59 °C. However, the reactions were performed under 100-120 °C. How to perform the reaction? Is there a safety problem? Furthermore, HFIP plays crucial roles in this transformation. However, only Re₂O₇ catalyst was investigated in HFIP. Other catalysts should also be investigated in HFIP. These results are valuable for this chemistry and the readers.

RESPONSE: The results for some common Bronsted acids are included in S2. Although HFIP has a boiling point of 59 °C, the reaction was performed in a sealed tube and heated with oil bath at the indicated temperature, similar practice has been widely adopted for reactions in HFIP even at higher temperature (140 °C: Chem. 2021, 7, 3425–3441; 100 °C: Angew. Chem. Int. Ed. 2017, 56, 3085–3089). Although we agreed that the bath temperature does not reflect the true reaction temperature inside the tube, lowering bath temperature often led to decreased reaction yields. Precautions were taken to minimize the potential safety issues and the experimental details are included in S4. Catalytic efficiency of various Bronsted acids as well as Lewis acids in HFIP has been provided in S2. For better comparison, the catalyst loadings were set to be the same as Re₂O₇.

- 2) **Comment:** (2) For the scope of substrate, there are several issues should be addressed. A) Could secondary and tertiary alcohols be applied in this transformation? B) Functional group tolerance should be investigated, such as carbonyl, ester, amino, etc. C) Why is the yield of product 2p low?

RESPONSE: A) For secondary and tertiary alcohols, the reactions are messy due to the fact that secondary and tertiary alcohols undergo dehydration easily under the reaction temperature, which often give a complicated mixture of alkenes and Friedel-Crafts products. However, this problem could be circumvented by put the secondary moiety on the ether part of the substrate with a primary alcohol, such as substrate **2qq**. Other catalysts (Bi(OTf)₃, Fe(OTf)₃) were also tested for secondary and tertiary alcohols, but only 10–20% of the desired C–O/C–O metathesis products were obtained. These results are mentioned in the text. B) Several new substrates were prepared and Substrates bearing phenolic OH (**2j**, **2rr**), carbonyl (**2o**, **2p**, **2q**, **2kk**), as well as phosphonyl (**2ll**) were well tolerated with different but predictable reactivity based on their electronic properties. Amino groups are not tolerated due to the fact they can quench the reactivity of Re₂O₇, however, protected amino groups such as sulfonamides (**2r**, **2z**) are well tolerated. C) Now **2p** is numbered as **2u** in this revised manuscript, we explained in details in **Figure 5** (after DFT calculation,

before discussion) for the low yield of this substrate, due to the fact of the formation of alkylated substrates, which could be minimized by changing the alkyl group on ether, yields could be improved to 77% (88% per cyclization) by switching Me group to a propyl group.

- 3) **Comment:** (3) *In order to compare, the same reaction conditions should be carried out for Fig. 5A and Fig. 5B.*

RESPONSE: Fig. 5A and Fig. 5B are two different sets of substrates, **1u** and **1u'**. We did not intend to compare Fig. 5A with Fig. 5B, but to compare **1u** with **1u'**, as well as **1b** with **1b'**. When the alkylation of substrate is the major reason for the low yield, the reaction yield can be increased by changing the alkyl group to a bulkier one (Figure 5A). However, when the alkylation of substrate is not the major reason, changing the alkyl group to a bulkier one only led to further decreased yield (Figure 5B).

- 4) **Comment:** (4) *For Supporting Information, the authors should check the data carefully. For examples, the reported ¹H NMR data in the experimental details section does not match the [spectrum/reported HRMS formula] for compound(s): 1m; the reported ¹³C NMR data in the experimental details section does not match the [spectrum/reported HRMS formula] for compound(s): 1g, 1h, 1q, 1r; HRMS data of products 2a-2k, 2m-2cc and 2ee-2ff were not attached, and relevant data were missing; the data for these new compounds need to be collated in SI.*

RESPONSE: Done, data has been double-checked and missing data were provided.

Reviewer #2:

General Comment: *In this manuscript, the authors report the Ring-closing C - O/C - O Metathesis of Ethers with Primary Aliphatic Alcohols leading to the synthesis of cyclic ethers. Similar transformations are already known for the synthesis of cyclic ethers via transesterification of two ether groups. However, this is the first example of cyclic ether synthesis via transesterification of alcohol and ether. But, the mechanistic investigation and anticipated novelty are insufficient for the proposed plausible mechanism and the authors should consider the following points:*

RESPONSE: As suggested by this reviewer, more experiments were carried out to elucidate the reaction mechanism. We outlined four possible pathways in **Fig. 2**, but specifically pointed out that **pathway (a)** was the operative one, this conclusion was

made based on various mechanistic experiments as well as DFT calculations.

- 1) Comment (1):** *C-O/C-O σ -bond metathesis strategy is already known for the synthesis of cyclic ethers that follows the activation of ethers and nucleophilic attack by another ether. But here authorS highlighted a different mechanism for metathesis that includes the activation of alcohol followed by a nucleophilic attack of less nucleophilic ethers which is new from the mechanistic point of view. Although it is the first transesterification between alcohol and ether, similar transformations are known.*

RESPONSE: Although C-O/C-O σ -bond metathesis strategy are already known for the synthesis of cyclic ethers via transesterification of two aliphatic ether groups (ref. 38–39), the reported protocols were only applied to aliphatic ethers (not working for aromatic ethers). We described in this manuscript the first example of cyclic ether synthesis via transesterification between alcohols and ethers, and successfully applied it to both aliphatic and aromatic ethers. From the mechanistic point of view, it is novel since transesterification between an alcohol and an ether was exclusively proposed to proceed via activation of ether (no matter aliphatic ether or aromatic ether, ref. 11–21). This new activation mode was then corroborated by a collection of mechanistic experiments as well as DFT calculations. Due to this new activation mode, our method is also applicable to substrates with multiple ether moieties, or functional groups that can compete for coordination of catalysts, which was problematic with previous methods.

- 2) Comment (2):** *Entry 6 of table 1 gave 10% yield in presence of DCE solvent so this reaction should be carried out in presence of fluorinated alcohol solvent.*

RESPONSE: Done, catalytic efficiency of various Bronsted acids as well as Lewis acids (including $\text{Fe}(\text{OTf})_3$) in HFIP has been provided in S2. For better comparison, the catalyst loadings were set to be the same as Re_2O_7 .

- 3) Comment (3):** *Optimization of the reaction condition has been carried out in presence of solvent DCE, TFE and HFIP but no other polar solvents have been screened with Lewis acids.*

RESPONSE: Other polar solvents such as ethanol, DMF, DMSO, etc. were also screened with Re_2O_7 and $\text{Fe}(\text{OTf})_3$, however, no desired product was detected, and that's why it is not included, some of this information is now included in S2.

- 4) Comment (4):** *No phenolic OH substrates have been screened in substrate scope. Is it possible to activate phenolic OH? What about the further possible nucleophilic attack by ether?.*

RESPONSE: Phenolic substrate **1a-2** give the cyclized product **2a** in significantly reduced yield (**Fig. S8**), and O¹⁸ labeling experiments showed that the product was formed by activation of ether followed by attack of phenolic OH, rather than OH activation followed by ether attack. More details are included in **Figure S16**.

- 5) **Comment (5):** *In substrate scope (Table-2) what will happen if allylic alcohols are used?. Will it undergo further cyclization?*

RESPONSE: For allylic alcohols, the reactions are messy due to the fact that allylic alcohols undergo dehydration easily under the reaction temperature, which often give a complicate mixture of alkenes and Friedel-Crafts products. Changing the solvents and/or lowering the reaction temperature did not led to any success either. This information is now included in the text.

- 6) **Comment (6):** *In mechanistic investigation (Figure-2) author has not given any evidence for the formation of perrhenate ester I.*

RESPONSE: In the initial manuscript, the formation of perrhenate ester I was proposed based on literature precedence and DFT calculations carried out in this work. We then tried to detect its formation by GC-MS, and it could be successfully detected when the reaction was performed at 30 °C, although no desired product was formed at this temperature. We also tried to detect it at standard reaction temperature, but with no success, probably because it quickly transformed to following intermediate and product. This information is now mentioned in text and included in **Figure S18** (supporting information **S5.6**).

- 7) **Comment (7):** *In mechanistic investigation (Figure-2) no separate experiment has been carried out for the identification of oxonium ion intermediate II and intermediate VI.*

RESPONSE: For intermediate II, we are unable to identify it at the reaction temperature, probably due to fleeting nature at the elevated temperature. Previous characterization of this intermediate was done at cryogenic conditions using an activated starting material (Ref .22–23). We proposed the involvement of intermediate II based on both DFT calculations as well as several mechanistic experiments, including O¹⁸ labeling experiments, which are all in good agreement with its formation. For intermediate VI, there might some misunderstanding, since this intermediate was proposed in Fig. 2 as potential alternative because analogous intermediate (ether activation) was proposed in reported transesterifications, however, our mechanistic experiments as well as DFT calculations helped us to rule out this possibility (please see the explanation in text for more details). As a result, intermediate VI could not be identified.

8) **Comment (8):** *The evidences in support of intermediates for plausible mechanism (Figure-2) are insufficient.*

RESPONSE: As an addition to response comment (8), we are sorry for this confusion, but we need to clarify that although four possible pathways (**a**, **b**, **c**, **d**) are proposed in **Fig.2** as potential mechanism. Mechanistic experiments as well as DFT calculations in the following section helped us to rule out **b**, **c**, **d**, and confirmed **pathway a** to be most likely operative pathway.

9) **Comment (9):** *The possible involvement of radical and other intermediates can not be ruled out.*

RESPONSE: We carried out a series of radical trapping experiments, however the reaction could proceed without problem and the corresponding radical adduct could not be detected. Additional DFT calculations (**Figure S25, S6**) were also performed to show that a radical pathway has much larger energetic span compared to **pathway a** in **Fig.2**. It has to be mentioned that the catalytic capability of Re_2O_7 is not tolerant of basic N atoms, so TEMPO is not a propriate radical trap. Re_2O_7 can also activate cyclopropane (a separate manuscript which is currently underway), so the design of a radical clock experiments is also problematic. More details about the radial trap experiments are provided in **Fig. 3E** and **S5.5**.

10) **Comment (10):** *Evidence for O-H activation (Figure-3a, 3b) is enough to conclude that reaction is initiated via O-H activation but what will happen if one tries to carry out the reaction with dual OH activation containing both aliphatic OH groups or one aliphatic and another phenolic OH group?*

RESPONSE: Substrate **1rr** containing one aliphatic and one phenolic OH group was synthesized, it selectively gave **2rr** as the major product, which resulted from C-O/C-O σ -bond metathesis between aliphatic OH and the ether. **2rr'** was obtained as a minor product, which resulted from activation of aliphatic OH followed by intramolecular displacement of phenolic OH (Derived by analogy to **1a-2** in **Fig.S8**, and **1d'''** in **Fig.S16**). This result is now included in **Table 2**.

11) **Comment (11):** *How the author was able to get ^{18}O incorporated substrate $1a''$ and $1d''$? A preparation procedure should be provided.*

RESPONSE: Detailed preparation procedures or all ^{18}O labeled substrates were provided in S5.4.

12) **Comment (12):** *In (Figure-3c) of entry1, $1a'$ was observed in 14% yield but the authors have not mentioned anything regarding this product in the optimization*

table. Maybe this is also a possible intermediate in a reaction that facilitates the C-O/C-O σ -bond metathesis.

RESPONSE: The alkylated substrate **1a'** is dead-end a side product, not an intermediate. In the optimization table, for entries where no product was formed, **1a'** was not detected either. As shown in **Fig. 3a**, when **1a'** was subjected to the standard reaction condition, no significant amount of product was formed. We speculated that the formation of **1a'** was due to the intermediate of II, attacked by another substrate of **1a**. This is corroborated by the results shown in **Fig. 3c**, increasing the size of alkyl groups, the formation of alkylated substrate was decreased, and the formation of product was increased. The potential reason for this was provided in the text. Based on this understanding, the yield could be enhanced by switching to a bulkier alkyl group for substrates where substrate alkylation was the major reason for low yields (**Fig. 5**). The relevant explanation was provided in the text as well.

REVIEWERS' COMMENTS

Reviewer #1 (Remarks to the Author):

The authors have addressed most points raised by this reviewer. Although the substrate scope of the method is rather limited, the reaction mechanism is new. Therefore, this reviewer recommend acceptance for publication in the light of following comments.

- 1) The same reaction conditions should be carried out for Fig. 5A and Fig. 5B.
- 2) The authors should check the numbers for the compounds in Fig. 5.

Reviewer #2 (Remarks to the Author):

Xie and coworkers have reported the ring-closing C–O/C–O Metathesis of Ethers with Primary Aliphatic Alcohols. Authors have addressed the comments raised by both reviewers satisfactorily. However, major shortcoming of the work is its novelty and the more challenging intermolecular version of the reaction is already known. Hence, this reviewer is concerned that this work may not meet the urgency criteria required for a communication. Though, this reviewer expect a higher standard for a paper in Nature communications, if Editor deems appropriate, the manuscript may be accepted.

RESPONSES TO THE REVIEWER'S COMMENTS:

Reviewer #1:

- 1) **Comment:** (1) *The same reaction conditions should be carried out for Fig. 5A and Fig. 5B.*

RESPONSE: Done, same reaction conditions were carried out for Fig.5A and Fig. 5B, and same conclusion could be made.

- 2) **Comment:** (2) *The authors should check the numbers for the compounds in Fig. 5.*

RESPONSE: Done, numbers were rechecked, and mistakes are corrected.

Reviewer #2:

General Comment: *Xie and coworkers have reported the ring-closing C - O/C - O Metathesis of Ethers with Primary Aliphatic Alcohols. Authors have addressed the comments raised by both reviewers satisfactorily. However, major shortcoming of the work is its novelty and the more challenging intermolecular version of the reaction is already known. Hence, this reviewer is concerned that this work may not meet the urgency criteria required for a communication. Though, this reviewer expect a higher standard for a paper in Nature communications, if Editor deems appropriate, the manuscript may be accepted*

RESPONSE: Although the more challenging intermolecular version of the reaction is already known, all previous protocols (intermolecular and intramolecular) proceeded via ether activation followed by alcohol attack that results in C-O/O-H metathesis (Fig.1C and Fig. 1D). Selective activation of alcohol rather than ether gave unusual C-O/C-O metathesis, and it is challenging under traditional conditions due to the stronger basicity of ether. As a result, from the mechanistic point of view, it is novel. Due to this new activation mode, our method is also applicable to substrates with multiple ether moieties, or functional groups that can compete for coordination of catalysts, which was problematic with previous methods as well.